# Current Evidence on the Association of Micronutrient Malnutrition with Mild Cognitive Impairment, Frailty, and Cognitive Frailty among Older Adults: A Scoping Review

**DOI:** 10.3390/ijerph192315722

**Published:** 2022-11-25

**Authors:** Norhayati Mustafa Khalid, Hasnah Haron, Suzana Shahar, Michael Fenech

**Affiliations:** 1Center for Healthy Aging and Wellness, Faculty of Health Sciences, Universiti Kebangsaan Malaysia, Jalan Raja Muda Abdul Aziz, Kuala Lumpur 50300, Malaysia; 2Genome Health Foundation, North Brighton, SA 5048, Australia

**Keywords:** scoping review, micronutrient, mild cognitive impairment, frailty, cognitive frailty, older adults

## Abstract

Micronutrient malnutrition is thought to play an important role in the cause of cognitive impairment and physical frailty. The purpose of this scoping review was to map current evidence on the association between micronutrient deficiency in blood and mild cognitive impairment, frailty, and cognitive frailty among older adults. The scoping review was conducted based on the 2005 methodological framework by Arksey and O’Malley. The search strategy for potential literature on micronutrient concentration in blood and cognitive frailty was retrieved based on the keywords using electronic databases (PubMed, Cochrane Library, Google Scholar, Ovid, and Science Direct) from January 2010 to December 2021. Gray literature was also included in the searches. A total of 4310 articles were retrieved and 43 articles were incorporated in the review. Findings revealed a trend of significant association between low levels of B vitamins (folate and vitamin B12), vitamin D, vitamin A, vitamin E, omega 3 fatty acid, and albumin, and high homocysteine levels in blood with an increased risk of mild cognitive impairment among older adults. The results also indicated that low vitamin D levels, albumin, and antioxidants (lutein and zeaxanthin) in blood were significantly associated with frailty among older adults, while β-cryptoxanthin and zeaxanthin in blood were inversely associated with the risk of cognitive frailty. Vitamin D and antioxidants seemed to be targeted nutrients for the prevention of cognitive frailty. In conclusion, a wide range of micronutrient deficiency was associated with either mild cognitive impairment or frailty; however, little evidence exists on the dual impairment, i.e., cognitive frailty. This scoping review can serve as preliminary evidence for the association between micronutrient deficiency in blood and mild cognitive impairment, frailty, and cognitive frailty among older adults and prove the relevancy of the topic for future systematic reviews.

## 1. Introduction

Cognitive frailty (CF) is a predementia syndrome characterized by the simultaneous presence of both frailty and cognitive impairment [1]. It has been discovered to be associated with disability and an increase in mortality rates [2]. Frailty is defined by five phenotype models: unintentional weight loss, fatigue, weakness, decreased gait speed, and physical inactivity [3]. Frailty is associated with the increased risk of falls, functional disability, hospitalization, poor quality of life, and death [3]. Mild cognitive impairment (MCI) is defined as a syndrome where cognitive decline is greater than is expected for an individual’s age and education level but does not notably interfere with daily life activities [4]. MCI is associated with an increased risk of dementia [4]. The prevalence rate of cognitive frailty is between 1.0% and 12.0% among community-dwelling older adults [4,5]. Frailty affects 12% of the community dwellers aged ≥50 years [6], while cognitive impairment is present at a range of 5.1% to 41% [7,8]. Furthermore, a recent systematic review concluded that physical frailty is associated cross-sectionally and prospectively with a worse cognitive trajectory among those with MCI [9].

Cognitive frailty can be influenced by several risk factors, including impaired cardiovascular function, unhealthy lifestyle habits (smoking and inadequate physical activity), poor nutritional status, and adverse psychosocial experiences [10]. Increased age, vitamin D deficiency, co-existence of depression, frailty, and declined functional mobility and processing speed are among the predictors of CF [8,11]. Furthermore, for frailty, the risk factors are increasing age, body fat, lower skeletal muscle, malnutrition, low calcium intake, abdominal obesity, and poor physical function [12,13]. On the other hand, the risk factors for mild cognitive impairment are advanced age, low education level, living alone, low level of life satisfaction, less engagement in mental activities, low intake of fruits and vegetables, and not practicing calorie restriction [8,14,15]. Adequate-to-high protein [16] and at least 16 micronutrients, i.e., beta-alanine, calcium, creatinine, fluorides, leucine, magnesium, omega-3 fatty acids, potassium, vitamin B6, vitamin B9 (folate), vitamin B12, vitamin C, vitamin D, vitamin E, vitamin K2, and zinc, have been reported to be beneficial in improving musculoskeletal health and/or cognitive function in older people [17]. However, quite often, nutritional deficiencies have not been given equal emphasis in the effort to improve health among the adult population [18].

Nutritional deficits in older adults can impair or affect their level of physical and cognitive functioning [19]. Specific nutrient deficiencies such as some B vitamins, minerals, lipids, and antioxidants that occur during the life cycle can intensify brain physiological mechanisms that may increase the vulnerability to DNA damage in brain cells, which contributes to cognitive decline [20]. Furthermore, micronutrient malnutrition such as vitamin D deficiency and low plasma levels of essential amino acids is also the cause of frailty [21,22]. Insufficiency of vitamin D [23], and low levels of n−3PUFA, in particular, docosahexaenoic acid (DHA) and eicosapentaenoic acid (EPA) [24], influence not only physical and musculoskeletal function but also cognitive functioning in older subjects.

Mild cognitive impairment, as well as physical and cognitive frailty are potentially reversible if detected early and can be subjected to appropriate intervention strategies at the primary stage; thus, older adults may be able to return to normal cognition [2,8]. Comprehensive multimodality interventions comprising of nutritional, physical, and cognitive intervention have been reported to be superior in reversing frailty in older adults [25]. A multidomain intervention which includes four intervention components such as diet, exercise, cognitive training, and vascular risk monitoring could improve or maintain the cognitive functioning of these individuals [26].

Although micronutrient deficiency has been reported to influence cognitive impairment, frailty, and cognitive frailty, to our knowledge, there is no comprehensive scoping review available regarding the association of micronutrient deficiency in blood with mild cognitive impairment, frailty, and cognitive frailty among older adults. For example, previous work has only highlighted the association between vitamins with MCI and AD [27]. Therefore, in the present study, we aimed to carry out a scoping review to map current evidence not only on vitamins but also the association of micronutrient deficiency in blood with a broader scope of cognitive disorders, namely, cognitive impairment, frailty, and cognitive frailty among older adults. This review may act as a precursor for a full systematic review and to improve the understanding of the association between micronutrient deficiency in blood and MCI, frailty, and CF, leading to personalized rehabilitation for reversing CF. Unlike systematic reviews and narrative reviews, scoping reviews evaluate the existing literature, summarize the findings, provide information on knowledge gaps, and are the first step before conducting a systematic review [28,29].

## 2. Materials and Methods

This study adopted the Arksey and O’Malley’s (2005) framework for scoping review and as the foundation of the methodology [28]. According to the framework, there are five different stages which include stage 1: identifying the research question; stage 2: identifying relevant studies; stage 3: selecting studies; stage 4: charting the data; and stage 5: collating, summarizing, and reporting the results.

### 2.1. Stage 1: Identifying the Research Question

Describing the definition of a relevant research question is a crucial step to define and refine the chosen research strategy. Based on the statement by Arksey and O’Malley, we have identified one research question that was used to guide our systematic search strategy and reporting of results [28]. The research question that we focused on was: is there any association between micronutrient concentration in blood with mild cognitive impairment, frailty, and cognitive frailty among older adults?

### 2.2. Stage 2: Identifying Relevant Studies

The identification of the relevant literature was performed using different resources. These included various electronic databases (PubMed, Cochrane Library, Google Scholar, Ovid, and Science Direct). Gray literature (i.e., unpublished or difficult-to-locate materials) searches of the local journal, Malaysian Nutrition Research Bibliography (2011–2014 and 2015–2017), and online local databases were also conducted to maximize the search activities.

Table 1 lists the initial keywords and search terms generated. The search terms from the keywords, subject headings, and synonyms such as blood micronutrient profiles, albumin, homocysteine, amino acids, minerals, vitamins, antioxidants, cognitive frailty, mild cognitive impairments, physical frailty, frailty syndrome, debility, older adults, aging population, older population, aging, and elderly were generated by the research team to capture any potential resources from the database. Boolean operators (AND, OR, NOT) were employed to combine the search terms using the related keywords. Table 2 shows the search strings generated.

### 2.3. Stage 3: Selecting Studies

Our review inclusion criteria were as follows: (1) articles written in the English language; (2) publications for the period between January 2010 and December 2021; (3) articles describing studies on the association of micronutrient profiles in blood and cognitive frailty among older adults; and (4) the definition of the older adult population by age according to the country of origin. A 10-year time frame was selected because it was expected that this period would cover the most significant results on the recent literature for this topic [30]. The following were the exclusion criteria: (1) articles concerning individuals of other age ranges instead of elderly subjects; (2) animal studies; and (3) review articles. The studies were chosen and recorded in accordance with the Preferred Reporting Items for Systematic Reviews and Meta-Analysis (PRISMA) flow diagram for the scoping review process [31]. The selection of the articles was performed in two stages. In the first stage, two researchers independently screened the titles and abstracts of all resources based on the inclusion criteria and search terms. Irrelevant abstracts were removed, and the researchers then retrieved full articles based on the abstracts that were chosen. The full articles were then independently screened by two researchers to identify items related to the objectives of the review in the second stage. Articles were removed if they were unrelated to the objectives of the review. The search results were managed using the Mendeley Desktop software.

### 2.4. Stage 4: Charting the Data

The data charting process generated a summary of the articles that corresponded to the study’s objectives and research question. Data were extracted from the selected articles by the researchers and included in the charting table. These data included information of the studies such as the author(s), year of publication, country origin, study design, participants’ characteristics, blood micronutrient profiles results and methods of measurement, and findings.

### 2.5. Stage 5: Collating, Summarizing, and Reporting the Results

The collating process of this scoping review followed the PRISMA flow diagram to ensure the review search results could be reported accurately [31]. Tables were created to summarize the findings of the selected articles and analyzed to elucidate the association of blood micronutrient profiles and cognitive frailty among older adults. The results of the extracted data were analyzed using descriptive statistics to provide a summary of the studies based on the number and types of studies and their research scopes.

## 3. Results

### 3.1. Characteristics and Participants of the Selected Studies

The identification step retrieved 4310 articles using the search engines mentioned and the gray literature. The duplicates were then removed, and the remaining 3591 articles were examined for relevant abstracts. This procedure resulted in 162 articles that were reviewed for inclusion eligibility. Finally, 43 articles that met the inclusion criteria were identified (Figure 1) with 31 studies that were conducted among older adults with mild cognitive impairments, and 11 studies on older adults with frailty (Table 3). There was also only one study on cognitive frailty as this concept was newly defined in 2013, and thus, related research was still limited. The majority of the studies were cross-sectional studies (*n* = 28), and the remaining were prospective (*n* = 10), retrospective (*n* = 2), and case-control (*n* = 3) studies. Cohort (prospective or retrospective study design) and case-control studies have specific advantages by offering a temporal dimension to measure disease incidents and their association with an exposure. With respect to the cross-sectional studies, it involved examining data on diseases and exposure at one particular time point; therefore, it could not assess the cause and effect relationship [32]. Studies were conducted in more than 23 countries, i.e., Italy, Chile, China, France, Switzerland, the United Kingdom, Sweden, Korea, the United States, Germany, Finland, Australia, Brazil, Japan, Malaysia, Thailand, India, Taiwan, Turkey, Mexico, Australia, Spain, and Sweden. The samples’ sizes ranged from 68 to 6257 participants. The participants’ ages in these studies ranged depending on the different methodology conducted.

### 3.2. Association of Vitamin D with Mild Cognitive Impairment, Frailty, and Cognitive Frailty

There were twenty-five studies that investigated the association between vitamin D with mild cognitive impairment and frailty (Table 3). From the identified studies, nine cross-sectional studies reported that lower vitamin D levels were significantly associated with mild cognitive impairment [33,34,35,36,37,38,39,40,41], while findings from eight cross-sectional studies [21,42,43,44,45,46,47,48] reported an association with frailty. Besides that, four prospective studies found that lower vitamin D levels were associated with mild cognitive impairment [49,50,51,52]; however, one prospective study conducted by Graf et al. (2014) found that vitamin D was not associated with mild cognitive impairment [53]. In regard to frailty, two prospective studies discovered that low levels of vitamin D were significantly associated with frailty [54,55], and one prospective study by Buta et al. (2016) revealed that low levels of vitamin D were not significantly associated with frailty incidence after the presence of cardiometabolic diseases was accounted for [56]. There were several techniques for measuring vitamin D identified in this review, including the immunoassay method [34,35,36,37,53] and the mass spectrometry method [47,51,52].

**Table 3 ijerph-19-15722-t003:** Summary of studies evaluating the association between vitamin D with mild cognitive impairment, frailty, and cognitive frailty involved in the elderly.

Author, Year, Country	Study Design	Participant Characteristics	Micronutrients in Blood Profiles’ Outcomes and Methods of Measurement	Findings
Chei et al., 2014 [33] China	Cross-sectional study	Subjects: 2004 older adults from the Chinese Longitudinal Healthy Longevity Survey (936 males and 1068 females) Age: 60 years old and older	Fasting venous blood was collected and then plasma was stored at −80 °C until analysis. Plasma 25(OH)D3 levels were measured using an enzyme-linked immunosorbent assay.	There was a reverse association between plasma 25(OH)D3 levels and cognitive impairment (OR = 2.15, 95% CI: 1.05–4.41, *p* < 0.05).
Chhetri et al., 2018 [34] France	Cross-sectional study	Subjects: 1680 older adults from the Multi-domain Alzheimer Disease Preventive Trial (MAPT) Age: 70 years old and older	Blood samples were taken during enrollment and total plasma 25- hydroxyvitamin (D3 and D2 forms) were measured using a commercially available electro-chemiluminescence competitive binding assay.	High vitamin D was associated with a reduced likelihood of physical limitation and cognitive impairment (OR = 0.97, 95% CI: 0.95–0.99, *p* = 0.011).
Hooshmand et al., 2014 [35] Sweden	Cross-sectional study	Subjects: 75 patients (29 with subjective cognitive impairment, 28 with mild cognitive impairment, 18 with AD) referred to the Memory Clinic Mean age: 61.6 years old	Plasma samples were obtained during the diagnostic workup. Plasma levels of 25(OH)D were determined using the DiaSorin immunoassay method.	Elevated plasma 25(OH)D was significantly associated with better cognitive status (OR = 0.969, 95% CI: 0.948–0.990 per increase of 1 nmol/L of 25(OH)D).
Lee et al., 2017 [36] Korea	Cross-sectional study	Subjects: 2940 older adults from the Korean Urban Rural Elderly cohort study Age: 65 years old and older	Blood samples were collected after an overnight fast and stored at −80 °C until the time of analysis. Serum 25-hydroxyvitamin D (25[OH]D) levels were measured using a chemiluminescence immunoassay.	Lower 25(OH)D levels were significantly associated with cognitive impairment (OR = 1.81, 95% CI: 1.11–2.94, *p* = 0.017).
Llewellyn et al., 2011 [37] United States	Cross-sectional study	Subjects: 3396 older adults from the Third National Health and Nutrition Examination Survey Age: 65 years old and older	Blood samples were collected, and serum 25(OH)D concentration was measured by radioimmunoassay.	Participants who were severely 25(OH)D deficient were more likely to suffer from cognitive impairment (OR = 3.9, 95% CI: 1.5–10.4, *p* = 0.02).
Pavlovic et al., 2018 [41] United States	Cross-sectional study	Subjects: 4358 patients from the Cooper Clinic in Dallas Age: 55–65 years old	Serum 25-hydroxyvitamin D (25(OH)D) concentration was measured by a DiaSorin liasion chemiluminescence analyzer.	Low vitamin D was shown to be significantly associated with cognitive impairment (OR = 1.24, 95% CI: 1.01–1.51, *P* = 0.038).
Rosa et al., 2019 [39] Brazil	Cross-sectional study	Subjects: 165 older adults Age: 80 years old and older	All blood samples were collected after a 12 h fast, and then the serum was stored at −20 °C until analysis. Serum levels of vitamin D (25-hydroxyvitamin D) were measured using chemiluminescent microparticle immunoassay on the BitLab system.	Low vitamin D (≤18 ng mL^−1^) was significantly associated with a high risk of cognitive decline. Older adults with vitamin D levels > 19 ng mL^−1^ showed a lower prevalence of cognitive decline (Prevalence ratio = 0.59, 95% CI: 0.39–0.87).
Sakuma et al., 2018 [40] Japan	Cross-sectional study	Subjects: 740 patients (527 males and 385 females from the Project in Sado for Total Health (PROST)) Age: 65 years old and older	Blood samples were collected at the time of enrollment. Serum 25(OH)D levels were measured by a double-antibody radioimmunoassay (RIA2).	Low serum 25(OH)D levels were independently associated with a higher prevalence of cognitive impairment (OR = 2.70, 95% CI: 1.38–5.28, *p* = 0.0110).
Vedak et al., 2015 [41] India	Cross-sectional study	Subjects: 86 older adults Age: 50 years old and older	Blood samples were collected by venipuncture after overnight fasting, and then the serum was separated and stored at −80 °C until analysis. Serum levels of 25(OH)D were measured using kits from Immunodiagnostic Systems.	Serum 25(OH)D levels showed a substantial positive correlation with cognitive domains, including attention, language, registration, and naming (*p* < 0.001), whereas, there was a moderately positive correlation with domains such as orientation, recall, remote memory, visuospatial, and verbal fluency (*p* < 0.001).
Chang et al., 2010 [42] Taiwan	Cross-sectional study	Subjects: 215 community-dwelling older adults (128 females, 87 males) Age: 65–79 years old	Fasting bloods were collected and the serum specimens were stored at −80 °C until analysis. Serum 25(OH)D was measured by DiaSorin 25-Hydroxyvitamin D 125I RIA.	Insufficient 25(OH)D status was strongly linked to frailty syndrome using the Fried Frailty Index (FFI) (OR = 10.74, 95% CI: 2.60–44.31).
Dokuzlar et al., 2017 [43] Turkey	Cross-sectional study	Subjects: 335 patients who attended geriatric polyclinics (88 frail, 156 prefrail, and controls) Age: 60 years old and older	Serum 25-hydroxy-vitamin D [25(OH)D] level was measured using the radioimmunoassay technique.	Level of 25(OH)D decreased as severity of frailty increased (*p* < 0.05).
Gutiérrez-Robledo et al., 2016 [21] Mexico	Cross-sectional study	Subjects: 331 community-dwelling older adults Age: 70 years old and older	Peripheral blood samples were drawn, processed, and stored at −70 °C until analyses. 25(OH)-vitamin D serum levels were measured by a commercially available enzyme-linked immunosorbent assay (ELISA).	Low 25(OH)-vitamin D levels were significantly associated with the probability of being frail as compared with those with sufficient vitamin D levels (OR = 8.95, 95% CI: 2.41–33.30).
Hirani et al., 2013 [44] Australia	Cross-sectional study	Subjects: 1659 community-dwelling older adults (males) Age: 70 years old and older	Fasting blood samples were collected from participants in the morning of their clinic visit. Serum 25D and 1,25D levels were measured by manual radioimmunoassay using single batch reagents.	Low serum 25-hydroxyvitamin D (OR = 2.66, 95% CI: 1.32–5.36, *p* = 0.006) and 1,25-dihydroxyvitamin D (OR = 1.86, 95% CI: 1.04–3.59, *p* = 0.04) levels were independently linked to frailty in older adults.
Pabst et al., 2015 [45] Germany	Cross-sectional study	Subjects: 940 older adults (478 males and 462 females) from KORA (Cooperative health research in the Region of Augsburg) Age study Age: 65–90 years old	Serum total 25(OH)D was measured using enhanced chemiluminescence immunoassay.	High levels of 25(OH)D were inversely associated with being prefrail or frail. The odds ratios (OR) with 95% confidence intervals (CI) were 0.52 (0.34–0.78) for levels of 15 to <20 ng/mL, 0.55 (0.37 to 0.81) for normal 25(OH)D levels of 20 to <30 ng/mL, and 0.32 (0.21 to 0.51) for serum levels in the highest range ≥30 ng/mL.
Alvarez-Ríos et al., 2015 [46] Spain	Cross-sectional study	Subjects: 592 participants Median age: 74 years old	Serum concentrations of 25(OH)D from fasting blood were determined by a fully automated immunoassay 149 electrochemiluminescence system.	Low levels of 25(OH)D were significantly associated with frailty (OR = 1.65, 95% CI: 1.02–2.67, *p* = 0.042).
O’Halloran et al., 2020 [47] Ireland	Cross-sectional study	Subjects: 4068 participants Age: 50 years old and older	Non-fasting whole blood samples were collected between 09:30 AM and 16:30 PM by venipuncture and the plasma was then separated from the blood samples within 48 h of collection and archived at −80 °C until assayed. Plasma 25-hydroxyvitamin D (25(OH)D) concentrations were quantified using liquid chromatography tandem mass spectrometry.	All 3 measures of frailty were associated with lower levels of vitamin D (relative risk ratios (RRRs) = 0.51–0.75).
Wilhelm-Leen et al., 2010 [48] United States	Cross-sectional study	Subjects: 5048 participants from the Third National Health and Nutrition Survey (NHANES III) Age: 60 years old and older	Blood samples were collected, processed, and stored at −70 °C until analysis. Serum 25-hydroxyvitamin D was measured using the DiaSorin radioimmune assay kit.	Low serum 25-hydroxyvitamin D concentrations were associated with frailty amongst older adults (OR = 3.7, 95% CI: 2.1–6.8 amongst white older adults and OR = 4.0, 95% CI: 1.7–9.2 amongst non-white older adults).
Graf et al., 2014 [53] Switzerland	Prospective study	Subjects: 428 inpatients from the Geneva geriatric hospital Age: 75 years old and older	The 25(OH)D level was performed from frozen plasma obtained at the day of inclusion in the study and stored at −80 °C. Participants were followed-up for two years. Plasma 25(OH)D levels were measured by electrochemiluminesence immunoassay using Cobas E601.	Vitamin D was not associated with cognitive status (RRRs = 0.96, 95% CI: 0.25–3.61, *p* = 0.948).
Granic et al., 2015 [49] United Kingdom	Prospective study	Subjects: 775 participants in the Newcastle 85 + Study Age: 85 years old and older	Serum 25(OH)D was obtained from fasting morning blood samples. Participants were followed-up at 1.5 and 3 years. Serum 25(OH)D was measured using the DiaSorin radioimmune assay kit.	Both low and high season-specific concentrations of 25(OH)D were linked with the increased risk of prevalent cognitive impairment (OR = 1.66, 95% CI: 1.06–2.60, *p* = 0.03; and OR = 1.62, 95% CI: 1.02 to 2.59, *p* = 0.04, respectively).
Matchar et al., 2016 [50] China	Prospective study	Subjects: 1202 cognitively intact older adults Age: 60 years old and older	Fasting venous blood was collected, processed, and stored at −80 °C until analysis. The mean follow-up duration was 2.0 ± 0.2 years. Plasma 25(OH)D3 was measured using an enzyme-linked immunoassay by Immunodiagnostic Systems Limited.	Low vitamin D levels were significantly associated with an increased risk of subsequent cognitive decline (OR = 2.0, 95% CI: 1.2–3.3) and impairment (OR = 3.2, 95% CI: 1.5–6.6).
Moon et al., 2015 [51] Korea	Prospective study	Subjects: 405 older adults from the Korean Longitudinal Study on Health and Aging (KLoSHA) Age: 65 years old and older	Serum 25(OH)D concentrations were measured with ultra HPLC–tandem mass spectrometry. Participants were followed-up for 5 years.	Severe vitamin D deficiency was independently associated with the future risk of MCI (hazard ratio (HR) = 7.13, 95% CI: 1.54–32.9, *p* = 0.012).
Slinin et al., 2012 [52] United States	Prospective study	Subjects: 6257 older adults (females) Age: 65 years old and older	Fasting morning blood was collected, processed, and stored at −70 °C until analysis. Participants were followed-up for 4 years, and 25(OH)D 2 (ergocalciferol) and 25(OH)D 3 (cholecalciferol) were measured using mass spectrometry.	Low 25(OH)D levels among older women were associated with a higher odd of global cognitive impairment (OR = 1.60, 95% CI: 1.05–2.42) and a higher risk of global cognitive decline (OR = 1.58, 95% CI: 1.12–2.22).
Buchebner et al., 2019 [54] Sweden	Prospective study	Subjects: 1044 community-dwelling women, aged 75 years with reassessments at ages 80 (*n* = 715) and 85 (*n* = 382) years	Non-fasting serum samples were collected and stored at −80 °C. Participants were followed for 10 years. Serum 25(OH)D was assayed using liquid chromatography-mass spectrophotometry linked to an HPLC system.	The 25(OH)D insufficiency was associated with increased frailty in age 75 and 80 years (0.23 vs. 0.18, *p* < 0.001; and 0.32 vs. 0.25, *p* = 0.001, respectively). No association between 25(OH)D and frailty was observed at age 85 years (0.38 vs. 0.34, *p* = 0.187).
Buta et al., 2016 [56] United States	Prospective study	Subjects: 369 women from the Women’s Health and Aging Study II Age: 70–79 years old at baseline	Serum 25-hydroxyvitamin D (25[OH]D) was measured using a radioreceptor assay. The mean duration of follow-up was 8.5 ± 3.7 years.	Low serum vitamin D concentration was associated with incident frailty in older women (HR = 2.77, 95% CI: 1.14–6.71, *p* = 0.02), but the relationship was not evident after accounting for the presence of cardiometabolic diseases (HR = 2.29, 95% CI: 0.92–5.69, *p* = 0.07).
Vogt et al., 2015 [55] Germany	Prospective study	Subjects: 727 older adults from KORA-Age study Age: 65 years old and older	Non-fasting blood samples were collected at baseline and stored at −80 °C until analysis. Participants were followed for 2.9 ± 0.1 years. Serum total 25(OH)D were measured using enhanced chemiluminescence immunoassay.	Participants with very low 25(OH)D levels (<15 ng/mL vs. ≥30 ng/mL) had significantly higher odds of prefrailty (OR = 2.43, 95% CI: 1.17–5.03) and prefrailty/frailty combined (OR = 2.53, 95% CI: 1.23–5.22), but not exclusively for frailty (OR = 2.63, 95% CI: 0.39–17.67).

### 3.3. Association of B Vitamins with Mild Cognitive Impairment, Frailty, and Cognitive Frailty

The association between B vitamins with mild cognitive impairment and frailty was reported in 10 studies (Table 4). Two cross-sectional studies reported that low folate levels were associated with an increased risk of mild cognitive impairment [57,58]. Furthermore, one retrospective, prospective study and case-control study, respectively, also found that low folate levels were linked to an increased risk of mild cognitive impairment [59,60,61]. Beyond that, two cross-sectional studies reported that low vitamin B12 levels were significantly associated with the risk of mild cognitive impairment [62,63], while one cross-sectional study by Soh et al. (2020) revealed that there was no significant association between low vitamin B12 and cognitive impairment [64]. In contrast, Rosa et al. (2019) found that high vitamin B12 levels were a risk factor for cognitive decline [36]. Other than that, one prospective and case-control study reported that increased homocysteine levels were associated with the risk of mild cognitive impairment [60,61]. In discussing frailty, one cross-sectional study by Dokuzlar et al. (2017) found that no association between vitamin B12 levels and frailty was evident [40]. The techniques that had been used for measuring serum or plasma vitamin B12 and folate included the immunoassay technique [57,59,60,61,62,63], vitamin kit [64], chemistry analyzer [43], and microbiological assays [47], whereas total plasma homocysteine was measured using the enzymatic assay [34] and immunoassay technique [60].

### 3.4. Association of Antioxidants with Mild Cognitive Impairment, Frailty, and Cognitive Frailty

There were six studies that investigated the association between antioxidants with mild cognitive impairment, frailty, and cognitive frailty (Table 5). One cross-sectional, prospective, and case-control study, respectively, found that low levels of vitamin E (tocopherol) were significantly associated with mild cognitive impairment [65,66,67]. Beyond that, one cross-sectional study by Mangialasche et al. (2013) reported that vitamin A deficiency was associated with an increased risk of mild cognitive impairment [68]. Furthermore, two cross-sectional studies also reported that low lutein and zeaxanthin were significantly associated with frailty [47,69]. However, only one cross-sectional study conducted by Rietman et al. (2019) reported that low levels of β-cryptoxanthin were significantly associated with the risk of cognitive frailty [69]. Serum levels of vitamins A and E were measured by high performance liquid chromatography (HPLC) [65,66,68].

### 3.5. Association of Protein with Mild Cognitive Impairment, Frailty, and Cognitive Frailty

The association between protein with mild cognitive impairment and frailty was reported in four studies, and all found that low serum albumin was significantly associated with mild cognitive impairment and frailty (Table 6). Two cross-sectional studies reported that low serum albumin was significantly associated with mild cognitive impairment [71,72]. Furthermore, one retrospective study conducted by Wang et al. (2018) among 1800 older adults, aged 60 years and older, revealed that low serum albumin was significantly associated with an increased risk of mild cognitive impairment [73]. Furthermore, one cross-sectional study by Dokuzlar et al. (2017) found that levels of albumin decreased as the severity of frailty increased (*p* < 0.05) [43]. Serum albumin was measured using a chemistry analyzer [71,72].

### 3.6. Association of Lipids with Mild Cognitive Impairment, Frailty, and Cognitive Frailty

There were three studies that investigated the association between lipids with mild cognitive impairment and frailty (Table 7). Low omega-3 index levels were significantly associated with mild cognitive impairment as reported in a cross-sectional study by Lukaschek et al. (2016) [74]. Other than that, one case-control study reported that the proportion of lower unsaturated fatty acids and higher saturated fatty acids were significantly associated with mild cognitive impairment [75]. Furthermore, Chhetri et al. (2018) also discovered that low n−3PUFA showed a higher likelihood of physical limitation [34]. Erythrocyte fatty acid composition was measured using gas chromatography [34,74,75].

## 4. Discussion

The purpose of the current scoping review was to map current evidence on the association of micronutrient deficiency in blood with cognitive impairment, frailty, and cognitive frailty among older adults. Figure 2 summarizes the findings on the association between micronutrient malnutrition with mild cognitive impairment, frailty, and cognitive frailty that was involved in the elderly. A previous review had shown that thirteen vitamins (i.e., vitamin A, thiamine, riboflavin, niacin, pantothenic acid, vitamin B6, biotin, folate, vitamin B12, vitamin C, vitamin D, vitamin E, and vitamin K) and three quasi-vitamins (i.e., choline, inositol, and carnitine) play a substantial role in ≥1 of 6 relevant pathways associated with Alzheimer’s disease (AD) [27]. Meanwhile, findings from this present review add on the evidence of the association between micronutrient malnutrition (i.e., vitamin D, B vitamins, antioxidants, protein, and lipids) and a broader scope of cognitive disorders, namely, cognitive impairment, frailty, and cognitive frailty among older adults. Comprehensive knowledge on the biomarkers that are specific to MCI, frailty, and cognitive frailty is essential to design appropriate preventive strategies. According to Orsitto et al. [76], older adults who were at risk of malnutrition were more likely to suffer from MCI. While Khater and Abouelezz [77] reported that MCI might be associated with nutritional risks in elderly patients.

In this review, the results from both the cross-sectional [33,34,35,36,37,38,39,40,41] and prospective [49,50,51,52] studies supported the evidence of lower vitamin D levels being significantly associated with mild cognitive impairment. For example, the cross-sectional study by Chei et al. among older adults aged 60 years and above showed that the risk of cognitive impairment was higher by 2.15 times among those with the lowest (5.7–31.6 nmol/L) vitamin D levels as compared to the highest (57.0–208.7 nmol/L) [33]. A prospective study involving 9704 women aged 65 years and older demonstrated that women with very low vitamin D (<25 nmol/L) levels had increased odds of global cognitive impairment at 1.60 times as compared with those who had sufficient vitamin D levels (≥75 nmol/L) [52]. Similarly, a meta-analysis by Etgen et al. also found that older adults with low vitamin D status showed an increased risk of cognitive impairment compared with normal vitamin D status by 2.39 times [78]. Furthermore, a double-blind, randomized, placebo-controlled trial study in China among older adults aged 65 years and older with MCI reported that vitamin D supplementation (800 IU/day) for 12 months appeared to improve cognitive function by reducing oxidative stress [79]. Older adults with cognitive impairment could have limited sunlight exposure as a result of living in nursing homes or being confined indoors, leading to lower vitamin D levels. However, even after adjusting these potential confounders, the odds ratio was statistically significant, emphasizing the robustness of the association between low plasma vitamin D and the increased odds of cognitive impairment [33].

There are various biological pathways that may describe the association between low vitamin D and mild cognitive impairment. First, the risk of cardiovascular disease, diabetes, and hypertension may be increased by vitamin D deficiency [80], and these conditions may in turn be correlated with cognitive impairment [81]. Secondly, vitamin D may play a role in brain detoxification pathways by reducing cellular calcium, inhibiting nitric oxide synthase synthesis, and protecting neurons from reactive oxygen species by increasing glutathione levels [82]. Thirdly, vitamin D induces neurogenesis and regulates the synthesis of neurotrophic factors that are essential for cell differentiation and survival [83]. As for the fourth association found, vitamin D is also an immunosuppressive agent and prevents autoimmune damage to the nervous system [82]. Lastly, vitamin D induces amyloid beta clearance and phagocytosis, and thus it protects against programmed cell death [84].

Folate is required for the synthesis of DNA and RNA nucleotides, the metabolism of amino acids, and the occurrence of methylation reactions in almost all tissues including the brain [85]. Results from three cross-sectional studies and one retrospective, one prospective, and a case-control study supported the evidence that low folate levels were significantly associated with an increased risk of mild cognitive impairment [57,58,59,60,61]. For example, a case-control study of individuals aged 60 years and older in China found that a higher serum folate level (>7 ng/mL) was associated with a lower risk of MCI (adjusted odds ratio 0.24, 95% CI: 0.11, 0.52, *p* = 0.000) [61]. The systematic and meta-analysis by Zhang et al. also supported that plasma/serum folate levels were lower in AD patients than those in controls with the standardized mean difference (SMD) at −0.60 (95% confidence interval (CI): −0.65, −0.55) [86]. Besides folate, vitamin B12 insufficiency is frequently linked to cognitive impairments [87]. Thus, results from the two cross-sectional studies supported the evidence of low vitamin B12 levels being significantly associated with the risk of mild cognitive impairment [62,63]. A systematic and meta-analysis of cross-sectional studies by Zhang et al. found higher levels of vitamin B12 (OR = 0.77, 95% CI = 0.61–0.97) being positive associated with better cognition, thus, it concurrently supported the current study [88]. Folate and vitamin B12 are required for homocysteine metabolism, and their deficiency results in elevated homocysteine concentration [89]. The results from one prospective and case-control study supported evidence that increased homocysteine levels were significantly associated with the risk of mild cognitive impairment [60,61]. A case-control study by Zhou et al. among older adults aged 60 years and older with and without MCI found that higher homocysteine levels were associated with the risk of MCI (*p* < 0.01) [61]. Our review findings were consistent with the previous meta-analysis that reported high homocysteine levels (risk ratios, RR = 1.93; 95% CI: 1.50–2.49) were particularly strong predictors of Alzheimer’s disease [90].

Possible mechanisms related to the association between folate and other B vitamins with mild cognitive impairment, particularly vitamin B12, play an important role in regulating gene expression and DNA synthesis, or specifically, as determinants of homocysteine detoxification and neurotransmitter synthesis [91]. Folate and vitamin B12 deficiencies can reduce the S-adenosyl methionine and increase plasma homocysteine, and these effects may lead to cognitive impairment through the oxidation of functional and structural neuron and endothelium [89], as well as the inhibition of methylation-dependent reactions that include synthesis of the neurotransmitter [92]. The possible mechanisms of homocysteine neurotoxicity involve disruptions in methylation and/or redox potential, thus stimulating calcium influx, irregular amyloid, and tau protein accumulation [93], which eventually can result in apoptotic events and neuronal death [94]. Elevated homocysteine contributes to oxidative stress and cerebrovascular and neurological lesions, cognitive and memory decline, thus playing a crucial role in the pathogenesis of neurodegenerative diseases [95].

Results from one cross-sectional, prospective, and case-control study support evidence of low levels of vitamin E (tocopherol) being significantly associated with mild cognitive impairment [65,66,67]. A population-based prospective study (CAIDE) among 140 older adults reported that the risk of cognitive impairment was lower among subjects in the middle tertile (>0.295 and <0.05 µmol/mmol) of the γ-tocopherol/cholesterol ratio than in those in the lowest tertile (≤0.295 µmol/mmol): the multi-adjusted odds ratio (OR) with 95% confidence interval (CI) at 0.27 (0.10–0.78) [66]. This finding, that is also supported by the systematic and meta-analysis of cohort studies, reported that some food intake was related with a decrease in dementia, such as vitamin E (RR: 0.80 95% CI: [0.65–0.9], *p* = 0.034) [96]. The mechanisms that may describe the association between low vitamin E and the risk of mild cognitive impairment are its role as an antioxidant in the human body that protects the central nervous system (CNS) from free radical-mediated damage [97]. Vitamin E also has biological properties such as anti-inflammatory activity and cell signal regulation which may be important for neuroprotection [98].

Results from one cross-sectional study supported the evidence that vitamin A deficiency was associated with an increased risk of mild cognitive impairment. Shahar et al. have found that one of the predictors of MCI was vitamin A deficiency (adjusted OR = 3.253; 95% CI = 0.972–10.886; *p* < 0.05) [68]. Wołoszynowska-Fraser et al. have made a review about vitamin A and retinoic acid in cognition and cognitive diseases and presented the conclusion that vitamin A and retinoic acid were essential for learning, memory, and other cognitive processes [99]. Possible mechanisms’ pathways associated with the relationship between serum vitamin A levels and cognitive function are described below. Firstly, antioxidation, anti-inflammatory, anti-cholinesterase, and memory-restorative functions have been proposed for all-trans retinoic acid in the carboxylic form known as vitamin A [100]. Secondly, retinoic acid is responsible for neuroimmunological functions and interacts with other signaling mechanisms regulated by the nuclear receptor. Thirdly, retinoic acid is associated with regeneration and cognition [101]. Lastly, vitamin A may be able to suppress the formation of beta-amyloid fibrin [102].

Results from two cross-sectional studies supported the evidence of low serum albumin being significantly associated with mild cognitive impairment [71,72]. Based on a national population-based study in England among 1752 older adults aged 65 years and older, it was reported that the odds ratios (95% confidence intervals) for cognitive impairment in the first (2.2–3.8 g/dL), second (3.9–4.0 g/dL), and third (4.1–4.3 g/dL) quartiles of serum albumin compared with the fourth quartile (4.4–5.3 g/dL) were 2.5 (1.3–5.1), 1.7 (0.9–3.5), and 1.5 (0.7–2.9) [71]. A previous systematic review also supported that low or decreasing serum albumin was a predictive factor of increased mortality [103]. Malnutrition has negative consequences for dementia patients, such as increased morbidity and mortality and decreased quality of life [104]. Several mechanisms have been proposed to explain the effects of serum albumin on patients with mild cognitive impairment risk. Firstly, serum albumin is correlated with the nutritional condition of older people [105], and persistent malnutrition plays a vital role in cognitive impairment [106]. Second, serum albumin is essential to maintain colloid osmotic pressure and blood volume in the human body [107]. Lower albumin levels interfere with the adequate delivery of blood to the central nervous system and can lead to cognitive impairment. Thirdly, oxidative stress injury is a non-negligible cognitive impairment pathogenic cause [108]. Serum albumin has a high antioxidant function [109], and a reduction in its amount contributes to the oxidation/antioxidation imbalance and cognitive impairment.

Results from one cross-sectional study supported the evidence that low omega-3 index levels were significantly associated with mild cognitive impairment [74]. A meta-analysis by Beydoun et al. has also found that higher intake of n-3 fatty acids as a protective factor vs. lower intake with RR was 95% CI: RR = 0.67(0.47,0.96) [90]. Several possible mechanisms have been proposed related to the association between low omega-3 and mild cognitive impairment risk. There is evidence that indicates that omega-3 polyunsaturated fatty acids (PUFA), eicosapentaenoic acid (EPA), and docosahexaenoic acid (DHA) may have neuroprotective properties [110]. Omega-3 PUFA levels in the brain appear to decrease with aging, indicating that low levels of EPA and DHA could lead to memory loss and other cognitive functions [111]. Both EPA and DHA may also have a role in synaptic plasticity, neurogenesis, cognition, and vascular health [112]. The benefits of DHA could be due to its derivative neuroprotection D1 NPD1, which helps to promote membrane fluidity, regulate apoptosis, and modulate inflammation [113]. Likewise, EPA plays a key role in the regulation of blood flow and inflammation [114].

According to a systematic review by Lorenzo-lopez et al. [115], malnutrition or poor nutrition may cause frailty or vice versa, which may cause malnutrition or poor nutrition among older adults. Other mediating factors that may have influenced the association between the two variables include poor dentition, swallowing problems, reduced sense of smell or taste, and deteriorating functional capacity. In this current review, results from eight cross-sectional studies supported the evidence that low levels of vitamin D were significantly associated with frailty [21,42,43,44,45,47,48]. Another two prospective studies supported that low levels of vitamin D were significantly associated with frailty [54,55]. The results of this review are in line with a meta-analysis by Zhou et al. that suggested low levels of vitamin D were significantly associated with the risk of frailty: OR of frailty for the lowest versus the highest level of vitamin D was 1.27 (95% CI = 1.17–1.38) [116]. A study conducted by Hirani et al. [44] among community-residing men reported that the association between frailty and both low 25D and 1,25D levels remained significant, even after adjustments for other measures such as self-reported health and a range of health conditions that could also affect the ability to spend time outdoors and access sun exposure. Two cross-sectional studies supported evidence that low lutein and zeaxanthin were significantly associated with frailty [47,69]. A systematic review made by Zupo et al. also reported that higher dietary and plasma levels of carotenoids (α-carotene, β-carotene, lutein, lycopene, β-cryptoxanthin, and total carotenoids) were found to reduce the odds of physical frailty [18]. One cross-sectional study supported evidence that a low level of albumin was significantly associated with the risk of frailty [43]. Our review findings were consistent with a meta-analysis by Mailliez et al. that concluded frail individuals had lower plasmatic albumin levels with standardized mean difference (SMD): −0.62 (−0.84;−0.41) than robust individuals [117]. Only one cross-sectional study supported evidence of low levels of β-cryptoxanthin that were significantly associated with the risk of cognitive frailty [69]. Previous systematic reviews highlighted the importance of considering carotenoids as biological markers to monitor micronutrient status and to adopt recommendations of increased fruit and vegetable intake for intervention purposes, prevention, and better management of disability among older adults [18].

There are several possible biological mechanisms that may describe the correlation between low vitamin D levels and frailty. Low vitamin D levels are associated with a higher risk of incident mobility limitation and disability [118] due to the effect of vitamin D on the metabolism of muscle cells, including the transport of calcium, inorganic phosphate absorption of energy-rich phosphate compounds, and protein synthesis [119]. Low vitamin D is related to an increased risk of Th1 cytokine-mediated autoimmune disorders that include rheumatoid arthritis [120]. Evidence also shows that inflammation plays a key role in the pathophysiology of frailty via an irregular, low-grade chronic inflammatory response [121].

Possible mechanisms associated with lutein and zeaxanthin and frailty risk are xanthophyll carotenoids that serve as important antioxidants and anti-inflammatory compounds [122], with a potentially protective role in chronic aging diseases that include bone health [123]. Carotenoid deficiency can lead to chronic diseases through multiple physiological systems as a central feature of frailty [47]. Previous research has reported that both inflammation and oxidative stress are correlated with frailty [124].

The mechanism that may explain the association between serum albumin and frailty risk is the indication of protein catabolism with the loss of muscle mass through low serum albumin. Additionally, low albumin levels could also suggest that systemic inflammation as previously mentioned plays a role in the development of frailty [125]. Low serum albumin can also imply the presence of high oxidative stress and its association with frailty, which further promotes the role of oxidative stress in frailty [126].

### 4.1. Research Gaps

A number of gaps in knowledge are apparent from this scoping review. Firstly, most of the studies that evaluated the association between blood micronutrient profiles, mild cognitive impairment, and frailty were cross-sectional studies [21,33,34,35,36,37,38,39,40,41,42,43,44,45,46,47,48,57,58,62,63,64,65,67,68,69,72,74,75]. In fact, case-control studies cannot demonstrate causality [61]. Hence, future prospective studies and randomized controlled trials need to be conducted to identify a potential direct causal relationship between micronutrient deficiency, mild cognitive impairment, and frailty. Neuroimaging and neuropathological studies at the molecular level would also be useful to identify the underlying mechanisms for these associations. Presently, we designed a randomized control trial study to determine the effectiveness of a multidomain intervention among older adults with cognitive frailty involving nutrition, psychosocial, physical activity, cognitive training, and management of metabolic and vascular risk factors modules [127].

Secondly, a small sample size of published studies may have affected the statistical power of the analyses produced [35,41,45,55,56,65,66,67,75]. Thus, a larger sample size is required in future studies to confirm the association between specific blood micronutrient deficiencies and mild cognitive impairment and/or frailty.

Thirdly, several studies were non-randomized; thus, these studies were not representative of the general population of older adults including patients from memory clinics, with the majority of subjects being farmers and housewives, a highly educated preventive medicine population, as well as hospital-based patients [35,38,40,52,58,65]. Therefore, future studies should also carefully consider participants’ settings in the study design, such as their ethnic group and geographical areas that would best represent the general population of older adults.

Lastly, there were also unmeasured confounding factors that might have affected the results, such as information on supplement intake, dietary factors, direct sun exposure, and physical activity [33,35,36,50,57,64]. Thus, future studies need to include these possible confounding factors because these confounding variables cannot be excluded.

### 4.2. Strengths and Limitations

This review is a current analysis of studies from 2010 to 2021 that investigated the association between micronutrient malnutrition with mild cognitive impairment, frailty, and cognitive frailty among older adults. However, the limitation of this review is that it only included articles that were published in English, and thus, other related research published in other languages may have been overlooked. Furthermore, this scoping review is a summary of publications with minimal-to-no analysis such as a meta-analysis. Hence, a future systematic review would prove the evidence on the association between micronutrient deficiency in blood and MCI, frailty, and CF.

## 5. Conclusions

This scoping review has highlighted the relationship between low levels of vitamin B12, vitamin A, omega fatty acid, vitamin D, and albumin and mild cognitive impairment based on published cross-sectional studies. However, the associations of folate, vitamin D, and homocysteine with mild cognitive impairment based on prospective studies are still limited. Furthermore, low levels of vitamin D, lutein, zeaxanthin, and albumin were significantly associated with frailty based on previous cross-sectional studies. In contrast, only one prospective study found that low levels of vitamin D were significantly associated with frailty. For cognitive frailty, low levels of β-cryptoxanthin and zeaxanthin were significantly associated with the risk of cognitive frailty based on only one cross-sectional study. The majority of the studies on the association between micronutrient malnutrition and mild cognitive impairment, frailty, and cognitive frailty in this review consisted of cross-sectional studies. Thus, future prospective studies and randomized controlled trials need to be conducted to identify a potential causal relationship between micronutrient deficiency and mild cognitive impairment, frailty, and cognitive frailty needs. Furthermore, several factors such as sample size calculation, participant settings, and possible confounding factors need to be considered when conducting future research.

## Figures and Tables

**Figure 1 ijerph-19-15722-f001:**
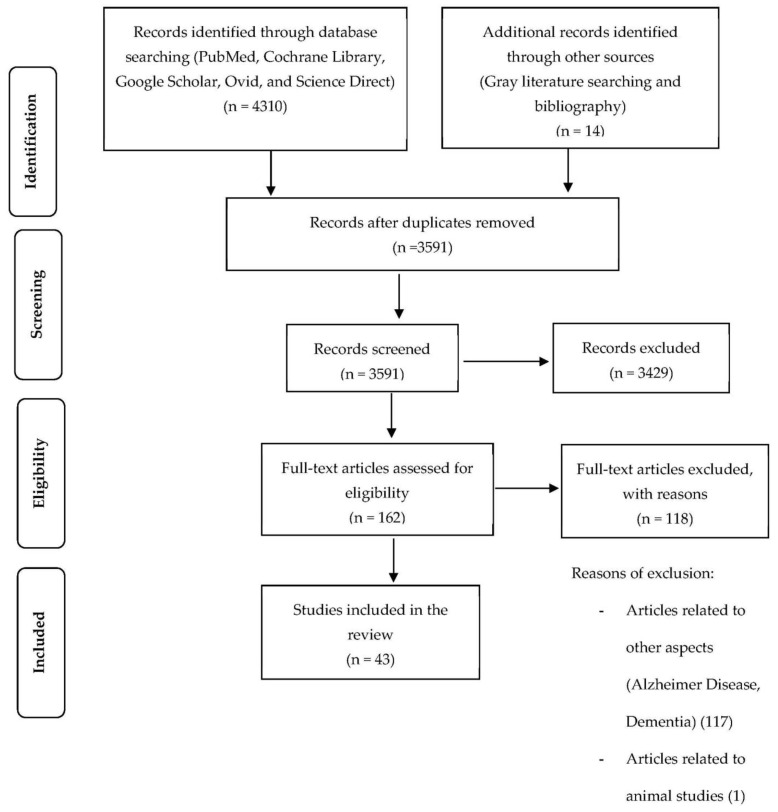
PRISMA flow diagram for the selection of related articles.

**Figure 2 ijerph-19-15722-f002:**
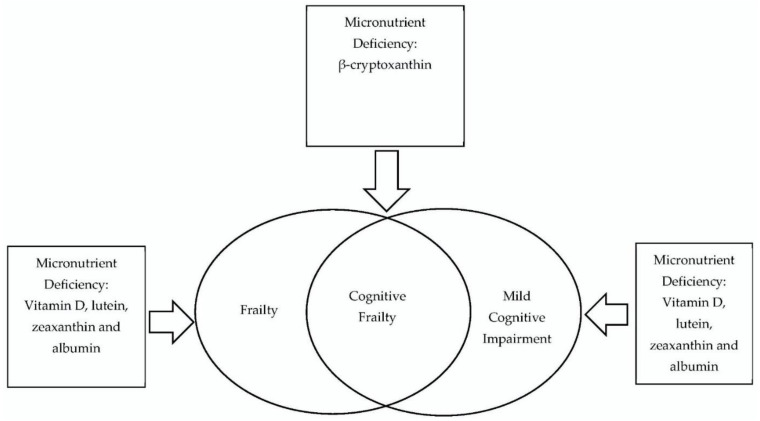
Summary of findings on the association between micronutrient malnutrition and mild cognitive impairment, frailty, and cognitive frailty involved in the elderly.

**Table 1 ijerph-19-15722-t001:** Lists of keywords and synonyms generated as search terms.

Blood Micronutrient Profiles	Cognitive Frailty	Older Adult
Albumin	Cognitive impairments	Aging population
Homocysteine	Cognitive dysfunction	Older population
Amino acids	Mild cognitive impairments	Aging
Fatty acids	Physical frailty	Elderly
Minerals	Frailty syndrome	
Vitamins	Debility	
Antioxidants		

**Table 2 ijerph-19-15722-t002:** List of search strings.

Search strings	“Blood micronutrient profiles” OR “Albumin” OR “Homocysteine” OR “Amino Acids” OR “Fatty Acids” OR “Minerals” OR “Vitamins” OR “Antioxidants” AND “Cognitive frailty” OR “Cognitive impairment” OR “Cognitive dysfunction” OR “Mild cognitive impairments” OR “Physical frailty” OR “Frailty syndrome” OR “Debility” AND “Older adults” OR “Aging population” OR “Older population” OR “Ageing” OR “Elderly”

**Table 4 ijerph-19-15722-t004:** Summary of studies evaluating the association between B vitamins with mild cognitive impairment, frailty, and cognitive frailty involved in the elderly.

Author, Year, Country	Study Design	Participant Characteristics	Micronutrient in Blood Profiles’ Outcomes and Methods of Measurement	Findings
Castillo-Lancellotti et al., 2014 [57] Chile	Cross-sectional study	Subjects: 1051 older adults who participated in the National Health Survey 2009–2010 Age: 65 years old and older	The blood samples were obtained from fasting individuals and were processed within 4 h of venepuncture. Serum folate and vitamin B12 levels were determined by competitive immunoassay using direct chemiluminescence.	Low serum folate levels significantly increased the risk of cognitive impairment (*p* = 0.026).
Kim et al., 2013 [58] South Korea	Cross-sectional study	Subjects: 321 older adults in the Songpa district Age: 60 years old and older	A fasting blood sample was drawn from each subject and the plasma samples were stored at −80 °C until analysis. Plasma folate and vitamin B12 were analyzed using a radioimmunoassay kit and a gamma counter. Plasma homocysteine was measured using HPLC with a fluorescence detector.	Plasma folate level was positively associated with the cognitive function test scores in the mild cognitive impairment (MCI) group (β = 0.840, *p* = 0.040).
Moore et al., 2014 [62] Australia	Cross-sectional study	Subjects: 1354 older adults from the Prospective Research in Memory (PRIME) clinics study and the Australian Imaging, Biomarkers and Lifestyle (AIBL) study and from patients attending for assessment or management of memory problems Age: 60 years old and older	Blood samples were taken within six months of cognitive testing. Serum vitamin B12 and red cell folate (RCF) levels were measured using the ADVIA Centaur chemiluminescent microparticle immunoassay, Tosoh immunoassay analyzer AIA600, the Roche Cobas 8000 electrochemiluminescence immunoassay, and Siemens Healthcare Diagnostic Immulite 2000 immunoassay.	Participants with low serum vitamin B12 (<250 pmol/L) and high red cell folate (>1594 nmol/L) levels were more likely to have impaired cognitive performance (adjusted odds ratio (AOR) = 3.45, 95% CI: 1.60–7.43, *p* = 0.002; and AOR = 1.74, 95% CI: 1.03–2.95, *p* = 0.04, respectively) as compared to participants with biochemical measurements that were within the normal range.
Rosa et al., 2019 [39] Brazil	Cross-sectional study	Subjects: 165 older adults Age: 80 years old and older	All blood samples were collected after a 12 h fast, and the serum was stored at −20 °C until analysis. Serum vitamin B12 was measured using the chemiluminescent microparticle immunoassay on the BitLab system.	A high concentration of B12 levels (≥496 pg mL^−1^) indicated a risk factor for cognitive decline (prevalence ratio = 1.90, 95% CI: 1.08–3.36).
Senger et al., 2019 [63] Brazil	Cross-sectional study	Subjects: 153 older adults Age: 80 years and older	Serum albumin levels were determined with the calorimetric method. Serum vitamin B12 was measured with the immunoassay technique.	Low vitamin B12 concentration positively associated with cognitive impairment (AOR = 5.37, 95% CI: 1.44–19.97, *p* = 0.012).
Soh et al., 2020 [64] Korea	Cross-sectional study	Subjects: 2991 older adults (1416 males and 1575 females) Age: 70–84 years old	Serum samples were collected, and vitamin B12 was measured with an Architect vitamin kit.	The association between the B12 group and cognitive function was not statistically significant (*p* > 0.05).
Dokuzlar et al., 2017 [43] Turkey	Cross-sectional study	Subjects: 335 patients who attended geriatric polyclinics (88 frail, 156 prefrail, and controls) Age: 60 years old and older	Serum vitamin B12 was measured using a diagnostic modular system autoanalyzer.	No association between vitamin B12 level and frailty (*p* > 0.05).
Mendonça et al., 2017 [60] United Kingdom	Prospective study	Subjects: 765 community-dwelling and institutionalized older adults Age: 85 years old and older	Blood was drawn between 7:00 and 10:30 AM after an overnight fast, and red blood cells (RBC) folate and plasma vitamin B12 were quantified by chemiluminescence. Total homocysteine (tHcy) was measured with an Abbot Imx immunoassay. Data were collected at baseline; 18 months, 36 months, and 60 months.	RBC folate and plasma tHcy were significantly associated with better global cognition at baseline (β = +1.02, SE = 0.43, *p* = 0.02; and β = −1.05, SE = 0.46, *p* = 0.02, respectively).
Zhou et al., 2020 [61] China	Case-control study	Subjects: 118 subjects with MCI and 118 subjects without MCI Age: 60 years old and older	Blood samples were collected, processed, and stored at −80 °C until analysis. The concentrations of serum folate and vitamin B12 were determined using the Abbott Architect-i2000SR automated chemiluminescence immunoassay system. The concentrations of plasma Hcy were determined by HPLC.	Increased Hcy levels and lower folate levels were independently associated with the risk of MCI (OR = 3.93, 95% CI: 1.54–10.07, *p* = 0.004; and OR = 0.24, 95% CI: 0.11–0.52, *p* = 0.000, respectively).
Baroni et al., 2019 [59] Italy	Retrospective	Subjects: 569 older adults attended the Centre for Diagnosis and Treatment of Cognitive Disorders (226 males and 343 females) Age: 60–96 years old	A 6-year observational, retrospective study was conducted by collecting routine blood analyses. Serum vitamin B12 and folate were measured by Immulite 2000 immunoassay system and plasma homocysteine by Cobas c702 chemistry analyzer.	Higher folate concentrations were significantly correlated with better cognitive performances (beta = 0.144, *p* = 0.001).

**Table 5 ijerph-19-15722-t005:** Summary of studies evaluating the association between antioxidants with mild cognitive impairment, frailty, and cognitive frailty involved in the elderly.

Author, Year, Country	Study Design	Participant Characteristics	Micronutrient in Blood Profiles’ Outcomes and Methods of Measurement	Findings
Kim et al., 2018 [65] Korea	Cross-sectional study	Subjects: 230 older adults from Yangpyeong cohort Age: 60–79 years old	Participants provided blood specimens after overnight fasting. Serum levels of vitamins A, C, and E (alpha tocopherol, beta tocopherol, and gamma tocopherol) were measured by high performance liquid chromatography (HPLC).	There was no significant association between the risk of cognitive impairment and serum levels of vitamin A and vitamin C (*p* > 0.05). β-gamma tocopherol levels were inversely associated with cognitive impairment (OR = 0.37, 95% CI: 0.14–0.98, *p* for trend = 0.051).
Shahar et al., 2013 [70] Malaysia	Cross-sectional study	Subjects: 333 older adults Age: 60 years old and older	Fasting venous blood was obtained, and the serum was stored at −40 °C until analysis. Vitamin A (serum retinol) and vitamin E (alpha tocopherol) statuses were determined in subsamples using HPLC.	Vitamin A deficiency was associated with an increased risk of MCI (AOR = 3.253, 95% CI: 0.972–10.886, *p* < 0.05).
O’Halloran et al., 2020 [47] Ireland	Cross-sectional study	Subjects: 4068 participants Age: 50 years old and older	Non-fasting whole blood samples were collected between 09:30 AM and 16:30 PM by venipuncture, and the plasma was separated from the blood samples within 48 h of collection and archived at −80 °C until assayed. Lutein and zeaxanthin were measured by the reverse-phase high performance liquid chromatography method.	All 3 measures of frailty were associated with lower levels of lutein (RRR = 0.43–0.63) and zeaxanthin (RRR = 0.49–0.63).
Rietman et al., 2019 [69] Europe	Cross-sectional study	Subjects: 2220 participants from Randomly Recruited Age-Stratified Individuals from the General Population (RASIG) study population Age: 35–74 years old	Analysis by the Universitaet Hohenheim.	Levels of β-cryptoxanthin and zeaxanthin were inversely associated with the risk of being cognitively frail (OR = 0.742, 95% CI: 0.604–0.911, *p* = 0.0043; and OR = 0.752, 95% CI: 0.588–0.960, *p* = 0.0225, respectively).
Mangialasche et al., 2013 [66] Finland	Prospective study	Subjects: 140 non-cognitively impaired older adult subjects derived from the Cardiovascular Risk Factors, Aging, and Dementia (CAIDE) study Age: 65–79 years old	Blood samples were taken after a minimum of 2 h fasting, and all serum samples were stored at −70 °C until analysis. The mean duration of follow-up was 8.2 years. Serum tocopherols, tocotrienols, αTQ, and 5-NO2-γ-tocopherol were measured with reverse-phase HPLC.	Elevated levels of tocopherol and tocotrienol forms were significantly associated with reduced risk of cognitive impairment in older adults (OR = 0.33, 95% CI: 0.11–0.97 for γ-tocopherol; OR = 0.21, 95% CI: 0.06–0.71 for β-tocotrienol; and OR = 0.33, 95% CI: 0.10–1.06 for γ-tocotrienol).
Yuan et al., 2016 [67] China	Case-control study	Subjects: 138 MCI patients and 138 age- and sex-matched healthy controls Age: 55–75 years old	Fasting venous blood samples were collected, processed, and stored at −80 °C until analysis. Plasma retinol and α-tocopherol were determined by using HPLC with UV detection.	Lower α-tocopherol was detected in the MCI patients compared to the control group (*p* < 0.05).

**Table 6 ijerph-19-15722-t006:** Summary of studies evaluating the association between protein with mild cognitive impairment, frailty, and cognitive frailty involved in the elderly.

Author, Year, Country	Study Design	Participant Characteristics	Micronutrient in Blood Profiles’ Outcomes and Methods of Measurement	Findings
Llewellyn et al., 2009 [71] United Kingdom	Cross-sectional study	Subjects: 1752 older adults (699 males and 1053 females) from the Health Survey for England 2000 Age: 65 years old and older	Non-fasting blood samples were collected, and serum albumin was measured using the DAX system.	Low serum albumin was independently associated with increased odds of cognitive impairment in the elderly population (OR = 2.5, 95% CI: 1.3–5.1, *p* for linear trend = 0.002).
Supasitthumrong et al., 2019 [72] Thailand	Cross-sectional study	Subjects: 182 older adults (60 with amnestic mild cognitive impairment (aMCI), 60 with Alzheimer’s disease (AD), and 62 normal controls (NC). Age: 55–90 years old	Fasting blood was collected between 8.00 and 8.30 AM, and serum albumin was measured using Architect C8000.	Serum albumin significantly affected cognitive functions (B = 0.148, SE = 0.061, *p* = 0.017), including episodic and semantic memory among elderly with MCI.
Dokuzlar et al., 2017 [43] Turkey	Cross-sectional study	Subjects: 335 patients who attended geriatric polyclinics (88 frail, 156 prefrail, and controls) Age: 60 years old and older	Serum albumin was measured using a diagnostic modular system autoanalyzer.	Level of albumin decreased as severity of frailty increased (*p* < 0.05).
Wang et al., 2018 [73] China	Retrospective study	Subjects: 1800 older adults Age: 60 years old and older	A 7-year retrospective cohort study was conducted by collecting data from medical records including serum levels of albumin. The method for measurement of serum albumin was not stated.	Low serum albumin levels at baseline (<40.5 g/L) were associated with the increased risk of MCI (HR = 2.18, 95% CI: 1.67–2.82).

**Table 7 ijerph-19-15722-t007:** Summary of studies evaluating the association between lipids with mild cognitive impairment, frailty, and cognitive frailty involved in the elderly.

Author, Year, Country	Study Design	Participant Characteristics	Micronutrient in Blood Profiles’ Outcomes and Methods of Measurement	Findings
Lukaschek et al., 2016 [74] Germany	Cross-sectional study	Subjects: 720 older adults from (Cooperative Health Research in the Region of Augsburg) KORA-Age study Age: 68–92 years old	Erythrocyte fatty acid composition was measured using gas chromatography.	Low omega-3 index levels were significantly associated with cognitive impairment (OR = 1.77, 95% CI: 1.14–2.76, *p* = 0.01).
Chhetri et al., 2018 [34] France	Cross-sectional study	Subjects: 1680 older adults from the Multi-domain Alzheimer Disease Preventive Trial (MAPT) Age: 70 years old and older	Blood samples were taken during enrollment. Erythrocyte membrane fatty acids were measured using gas chromatography.	Low n−3PUFA showed higher likelihood of physical limitation (OR = 1.55, 95% CI: 1.12 to 2.15, *p* = 0.009).
Yuan et al., 2016 [75] China	Case-control study	Subjects: 60 MCI subjects and 60 age- and gender-matched control adults Age: 55 years old and older	Fasting venous blood samples were collected between 8:00 and 9:00 AM from each subject. A fatty acid analysis was performed using gas chromatography (GC).	Lower erythrocyte unsaturated fatty acid and higher saturated fatty acid proportions might predict cognitive function decline in elderly Chinese adults. The percentage of erythrocyte DHA was positively correlated with the total MoCA score (*r* = 0.356, *p* < 0.05), while 12:0 fatty acid was inversely associated with the total MoCA score (*r* = 0.450, *p* < 0.05).

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
