# Peer review of "Current Evidence on the Association of Micronutrient Malnutrition with Mild Cognitive Impairment, Frailty, and Cognitive Frailty among Older Adults: A Scoping Review"

_ijerph, 2022, doi:10.3390/ijerph192315722_

Round 1
Reviewer 1 Report (Previous Reviewer 3)
No further comments - the Authors have addressed the raised issues.
Author Response
Dear Reviewer,
Thank you for reviewing our manuscript.
Reviewer 2 Report (New Reviewer)
Dear Editor,
I appreciate the opportunity to review the article "Current Evidence on the Association of Micronutrient Malnutrition with Mild Cognitive Impairment, Frailty and Cognitive Frailty among Older Adults: A Scoping 4 Review "
Within the methodological choices it would be important to explain why the research was conducted in the last 10 years , what is the explanation for this cut-off.
The abstract mentions these main results: Findings revealed a significant association between low levels of B vitamins (folate and vitamin B12), vitamin D, vitamin A, vitamin E, omega 3 fatty acid and albumin, and high homocysteine level in blood with increased risk of mild cognitive impairment among older adults. Can you get these data and these objectives with this scoping review type metdology? Wouldn't it be better to obtain it through a systematic review study?
In the methodology it mentions as a starting question "Is there any association between micronutrient concentration in blood with mild cognitive impairment, frailty, and cognitive frailty among older adults?" This question is not a question for a scoping review where the goal will be to map what exists on a given topic.
In table 3 it would be important to put the study design and statistical data of the results.
This article could be presented in another methodology with greater publication impact through a systematic review with metaanalysis.
I suggest revising the entire article according to this type of methodology.
Round 2
Reviewer 2 Report (New Reviewer)
Dear Editor,
I appreciate the opportunity to review this article.
I would like thank the authors for the revisions that have improved the quality of the article.
This manuscript is a resubmission of an earlier submission. The following is a list of the peer review reports and author responses from that submission.
Round 1
Reviewer 1 Report
General comments.
The challenge for any peer reviewer when it comes to this kind of review article is that it is very difficult to criticize the information itself. The manuscript reminds me of an introductory paragraph that many graduate students write to start their thesis. It is a broad summary of almost disparate detail presented with fact after fact after fact. It is extremely difficult to digest this kind of manuscript. Perhaps this kind of review is something that is necessary because it attempts to be exhaustive in summarizing existing knowledge. But frankly, what I came away with after reading this manuscript was probably little more than I could have achieved in a similar amount of time by searching Pubmed, and reading abstracts.
The main value for others interested in the field is this kind of review allows them to double-check whether they might have missed pertinent literature in their own review of the subject matter.
Specific comments
1. Line 182, and Table 3. The authors make no distinction between cross-sectional, prospective, retrospective or case-control studies. All studies are presented together as though there were no qualitative difference between them. Everything is presented as a superficial summary of the literature available.
2. Table 3. Hirani et al. This does not tell what direction the associations were in. It is not enough to say there was an association.
3. Table 3. Please separate cross-sectional, and prospective studies. Provide more detail about when the serum sample was taken for the nutrient assay. Was the timing months or years before the followup?
4. From line 273. Discussion. The discussion seems to read like a results section, and offers minimal actual discussion or insight. For example, is more research really needed or is that current status convincing? What could change if there were more such research?
5. Missing from the discussion is mention of the obvious causes of relationships observed with development of frailty:
A. Might frailty and cognitive impairment not cause nutritional changes resulting in different nutrient levels measurable in the blood?
B. Might vitamin D differences not be attributable to less outdoor activity and less sun exposure on the part of frail or those in mental decline?
C. Are the nutrient differences with frailty versus control subjects and impairment vs control, a cause or a consequence of frailty and impairment?
6. Line 483. It is my impression from reading this manuscript and some others cited in it, that NO more such studies are needed. The statement by the authors that more studies are needed seems to repeat a mantra that is often seen when research is summarized. At what point is it appropriate to say that enough research on a topic has been done?
7. Line 502. Limitations. I disagree that this "review is a current analysis of the association... " The limitation is that this is a summary of publications, with minimal/no analysis. This is not a meta-analysis.
Reviewer 2 Report
I was torn between a 3 and 4 for the overall rating and organization, so gave a 3 in one case and a 4 in the other. The paper is a "scoping review". The authors need to explain the difference between a scoping review and a general review more clearly. They also need to describe what is meant by "grey literature. They also need to be clear when they are talking about physical frailty, which they often just call frailty, and when they are talking about cognitive frailty. They only had one study on cognitive frailty but 13 on MCI and I assume there must have been subjects with cognitive frailty in other studies than the one they cited; this could have been discussed more fully. The results could have been summarized in a table both for cognitive frailty and frailty. Although I checked minor revisions, I do think summary tables would be a strong addition
Reviewer 3 Report
The Paper presents an important summary of the current cross-sectional associations between micronutrients and cognitive impairment, including frailty.
Following is of concern:
1. English language editing would aid in overall readability. Certain sentence constructions sound odd and are tough to understand.
2. A figure describing the main findings of the scoping review would help in understanding the main directions for future research.
3. The conclusion would benefit from stating so far observed associations in more detail (direction, strength, etc.), instead of simply outlining which relationships have been found so far.
4. Do the Authors plan a particular nutritional intervention or a cohort study in an elderly population?
Round 2
Reviewer 1 Report
General comments
There is so much going on in this manuscript that it is impossible to know where to start. The topic is an interesting one, but the subject area has already been reviewed many times before. Frankly, as this reviewer mentioned before, the present manuscript reads more like an introductory chapter of a graduate student thesis. It covers a lot and is presented as a mishmash. But with a thesis, subsequent chapters should fill in many of the chapter’s shortcomings. What is missing in the present manuscript are real critiques and conclusions, such as those previously published by one of the authors (Fenech 2017).
Simply insisting in the rebuttals that this is a “scoping review” does not justify the fact that this manuscript reads simply a long listing of what reads random facts pulled from the literature.
Many of the authors’ rebuttals to reviewer comments were as quoted here, “Already stated in the discussion line 1804.” BUT the manuscript only has 615 lines of text. What is meant by “Already stated”??? The many such author rebuttal statements to the editor and reviewers, and they have absolutely no basis of evidence in the text.
Specific Comments
1. The authors need to address earlier similar work; in particular, discuss the excellent publication written by one of the present authors (Fenech 2017) . Explain why the present review is better, different or necessary, compared to earlier similar publications. Oddly, the present submission does not even look as though it is the work of one of the authors, Fenech, other than including him as the senior author.
2. In my previous review I suggested that the authors separate the categories of study designs they are summarizing in their Table 3, and this revision does that. But does a reviewer need to point out everything? Table 3 continues to lack any logical organization: research involving folic acid is randomly mixed in among the protein, or vitamin D, or homocysteine or antioxidants or B vitamins. There is no organization here. Far better would be to address each nutrient of interest with a separate table. Or perhaps each nutrient as a separate manuscript for publication.
3. Most author rebuttals make no sense. For example: Previous reviewer comment #6 was rebutted with “Already stated in line 98 and line 2023.” Well, this version’s Line 98 has nothing to do with the comment, and Line 2023 does not exist.
PLEASE CITE AND ADDRESS THE FOLLOWING. WHAT DOES THE PRESENT MANUSCRIPT OFFER THE READER THAT FENECH DOES NOT?
Fenech, Michael. 2017. “Vitamins Associated with Brain Aging, Mild Cognitive Impairment, and Alzheimer Disease: Biomarkers, Epidemiological and Experimental Evidence, Plausible Mechanisms, and Knowledge Gaps.” Advances in Nutrition 8 (6): 958–70. https://doi.org/10.3945/an.117.015610.